# Ghost in the Minecraft: Hierarchical Agents for Minecraft via Large Language Models with Text-based Knowledge and Memory

## Abstract

As modern computer games continue to evolve, there is a growing need for adaptive agents that can effectively navigate, make decisions, and interact within vast, ever-changing worlds. While recently developed agents based on Large Language Models (LLMs) show promise in adaptability for controlled text environments, expansive and dynamic open worlds like Minecraft still pose challenges for their performance. To address this, we introduce *Ghost in the Minecraft* (GITM), a novel hierarchical agent that integrates LLMs with text-based knowledge and memory. Structured actions are constructed to enable LLMs to interact in Minecraft using textual descriptions, bridging the gap between desired agent behaviors and LLM limitations. The hierarchical agent then decomposes goals into sub-goals, actions, and operations by leveraging text knowledge and memory. A text-based in-context learning method is also designed to enhance future planning. GITM demonstrates the potential of LLMs in Minecraft's evolving open world. Notable milestones are collecting 99.2% of items and a 55% success rate on the popular "ObtainDiamond" task. GITM also shows impressive learning efficiency, requiring minimal computational resources.

## 1 Introduction

In modern computer games, NPCs (non-player characters) are more than just background decoration or task distributors (Laird and VanLent, 2001; Yannakakis, 2012). They have their own lives, needs, and daily routines in the game world. However, current NPCs, whether based on traditional designs such as Finite State Machines (FSMs) (Champandard, 2003) and Behavior Trees (Palma et al., 2011), or on model-driven approaches such as Reinforcement Learning (RL) (Wang et al., 2009; Makri and Charalambous, 2021), still struggle to adapt to changing game environments. This results in NPCs that often exhibit illogical, awkward, and irrational behavior when encountering unexpected game scenarios or changes. The lack of adaptability not only diminishes the realism and immersion in games, but also highlights a broader issue: the need for agents capable of autonomously navigating, interacting with, and executing tasks within constantly changing environments.

The recent emergence of Generative Agents (Park et al., 2023) based on Large Language Models (LLM-based agents) has attracted much attention. This innovative approach empowers agents to mimic daily human activities like waking up, making breakfast, working, and even creating art and writing - all within a controlled, text-based gaming environment called Smallville (Park et al., 2023). While these generative agents demonstrate strength in mimicking believable human activities, Smallville itself offers only a narrow, controlled game environment. In this limited setting, agents act within pre-defined roles and behaviors specific to crafted scenarios and tasks. The adaptability of LLM-based agents in more open and dynamic game worlds remains an open problem.

Developing an agent that can navigate, make decisions, and interact effectively in a vast and ever-changing world presents unique challenges. However, the highly controlled environment of Smallville obscures such complexities and uncertainties. To delve deeper into this challenge, we chose Minecraft as our experimental platform due to its massive scale, vast landscape, and unrestricted freedom. Minecraft introduces procedurally generated terrain, where agent interactions also have a persistent impact on the environment, requiring agents to handle unpredictable situations and adapt to diverse

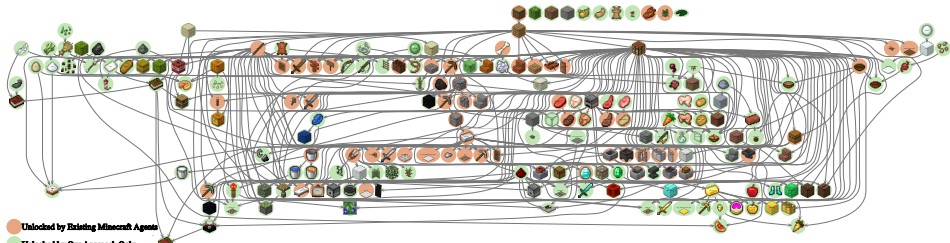

Figure 1: **GITM unlocks 99.2% of the technology tree in Minecraft.** Each node represents an individual item. The directed edges between nodes represent prerequisite relationships for obtaining. For better readability, we manually merge some similar nodes, *e.g.*, "wooden_pickaxe/axe/hoe/shovel" are merged into one node, and "wooden_pickaxe" is selected to represent the merged node. Existing Minecraft agents Baker et al. (2022); Hafner et al. (2023); Wang et al. (2023) only unlocked 78 / 262 = 30% items, while our GITM successfully unlocked 260 / 262 = 99.2% items.

surroundings. Researchers must take into account the diversity and unpredictability of Minecraft, requiring innovative solutions to ensure high adaptability.

Enabling LLM-based agents to physically interact within the Minecraft world using keyboard and mouse controls presents a major challenge. This is because LLMs inherently lack intuitive mastery of these controls. To overcome this limitation, we develop a structured action set that translates common in-game physical interactions into executable textual descriptions. This approach bridges the gap between desired agent behaviors and LLMs' inability to fluidly handle physical in-game interactions. With this action set translating inputs into text, LLMs can now leverage their language skills to participate in Minecraft's physical game world. This allows the LLM to make decisions in an abstract textual space rather than directly interacting with the physical environment. However, this conversion is not perfect, as it cannot fully capture the complexity of tasks and the uncertainty of physical interactions. To better address these challenges, we developed a hierarchical agent that starts with an overall goal, breaks it down into sub-goals, structured actions, and finally specific keyboard/mouse operations. We also design a text-based in-context learning strategy that utilizes text-based knowledge and memory to improve the accuracy of LLM in physical interaction.

Specifically, we first construct a set of structured actions by LLM through action decomposition and clustering from 3141 pre-defined MineDojo (Fan et al., 2022) tasks. The structured actions are defined with clear semantics and corresponding feedback, enabling LLM to understand surrounding environments and make decisions at the cognitive level. LLM can use them to physically interact with the environment and complete various tasks. Then, the agent uses a hierarchical structure with an LLM decomposer, planner, and interface module to break down goals into sub-goals, structured actions, and executable operations. When given a goal, the decomposer breaks it down into a series of sub-goals based on relevant text-based knowledge gathered from the internet. The planner then maps out a sequence of structured actions to accomplish each sub-goal. Finally, the LLM Interface module executes these planned actions to interact with the environment. It does this by processing raw keyboard/mouse input and observations. We also design a text-based in-context learning strategy that records and summarizes successful action lists into a text-based memory to enhance future planning.

In this paper, we demonstrate the feasibility of developing autonomous agents in Minecraft using Large Language Models (LLMs). LLM can leverage its vast knowledge and reasoning capabilities to provide logical responses to unforeseen or out-of-scope situations. It can also rapidly enhance its interaction capabilities and goal completion using text-based knowledge and memory. Our approach, Ghost In the Minecraft (GITM), showcases the potential for autonomous agents to address a wide range of challenges in vast and ever-changing environments, allowing them to effectively navigate such open-world settings. As a result, our agent has successfully collected 99.2% items in the Minecraft Overworld as a milestone (see Fig. 1). It also achieves a decent success rate of 55% for the popular "ObtainDiamond" task. Our agent demonstrates superior learning efficiency. It could be trained with 32 CPU cores in just two days.

## 2 RELATED WORK

**Minecraft agents** are intelligent programs that can perform various tasks within the Minecraft world. Reinforcement learning has dominated this area for many years. Some initial attempts have tried to use hierarchical RL (Skrynnik et al., 2021; Mao et al., 2022; Lin et al., 2021) or imitation

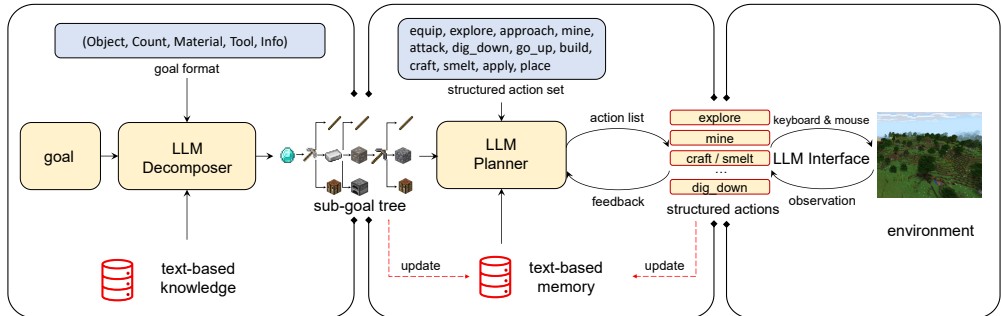

Figure 2: **Overview of our GITM.** Given a Minecraft goal, the LLM Decomposer divides the goal into a sub-goal tree. The LLM Planner then plans an action sequence for each sub-goal. Finally, the LLM Interface executes each action in the environment. Our LLM-based agents can be further enhanced by leveraging text-based knowledge and memory.

learning (Amiranashvili et al., 2020) in MineRL competitions (Milani et al., 2020; Guss et al., 2021; Kanervisto et al., 2022). Recently, with large-scale web data, VPT (Baker et al., 2022) builds a foundation model for Minecraft by learning from videos. Based on its success, many works (Milani et al., 2023) have also explored to fine-tune foundation model with human feedback. On the other hand, as Minecraft agents become increasingly proficient in handling simple tasks, the importance of multi-task learning becomes more prominent. Some previous works have adopted knowledge distillation (Tessler et al., 2017) and curriculum learning (Kanitscheider et al., 2021), while recent works (Fan et al., 2022; Cai et al., 2023) tried to construct a language-conditioned multi-task agent via feeding the goal description embedding into the model.

Recently, researchers have come to be aware of the extraordinary general planning ability of LLMs (Huang et al., 2022a). Many works (Huang et al., 2022b; Wang et al., 2023; Yuan et al., 2023) have leveraged LLMs to enhance the high-level planning ability of Minecraft agents. Inner Monologue (Huang et al., 2022b) leveraged environment feedback to improve the planning ability of LLM. DEPS (Wang et al., 2023) further extended this closed-loop interaction by introducing a description, explainer, and selector. Plan4MC (Yuan et al., 2023) pre-defined basic skills and instructed LLM to extract the relationship between skills to construct a skill graph.

Unlike previous RL-based or RL with LLM methods, our LLM-native approach brings the Minecraft agent to another level both in efficiency and robustness by leveraging high-level action abstraction and text-based knowledge and memory.

**Large Language Models with Tools** Extending the ability of LLMs by leveraging external tools has drawn a lot of attention recently. Several works (Liang et al., 2022; Singh et al., 2022; Driess et al., 2023) have explored to augment LLMs with robot perception and control abilities. Code as Polices (Liang et al., 2022) tried to prompt LLM to generate codes that can drive robots. PaLM-E (Driess et al., 2023) unified robot perception, instruction following, task planning, and low-level control into a unified framework. Another line of works tries to build external plugins around LLMs to enhance its ability. Toolformer (Schick et al., 2023) tries to teach LLMs to choose and use a wide range of tools like calculators and search engines and incorporate the results from tools into text generation. HuggingGPT (Shen et al., 2023) builds an agent for leveraging a combination of vision, language, and audio models hosted on HuggingFace. API Bank (Li et al., 2023) proposes a synthetic benchmark suite for evaluating how good LLMs are for using external tools.

Compared with these tool-augmented LLMs, our agents are tasked with much more complex goals in a highly uncertain open-world.

## 3 METHOD

### 3.1 ACTION ABSTRATION

LLM is not proficient at handling physical interactions based on keyboard and mouse operations, therefore we abstract the actions into a set of structured actions represented in text. The structured actions are designed with well-defined functions and clear semantics, enabling LLMs to make decisions at the cognitive level. A structured action can be defined as follows:

$$(\texttt{ActionName, Arguments, Description}), \tag{1}$$

Table 1: **Examples of structured actions.** A structured action contains name and arguments for execution, as well as description to help LLMs understand and decide when to choose this action.

| Name | Arguments | Description |
|---|---|---|
| equip | object | Equip the object from the inventory: used to equip equipment, including tools, weapons, and armor. |
| explore | object, strategy | Move around to find the object: used to find objects including block items and entities on the ground. |
| approach | object | Move close to a visible object: used to approach the object you want to attack or mine. |
| mine/attack | object, tool | Attack/Mine the object with the tool: used to attack/mine the object within reach. |
| dig_down/go_up | ylevel, tool | Dig down/Go up with the tool: used to go down/up underground. |
| craft/smelt | object, tool, material | Craft/Smelt the object with the materials and tool: used to craft new object that is not in the inventory or is not enough. |
| apply/place | object, tool | Apply/Place the tool on the object: used to apply tools or place blocks. |

The action name and arguments define the action we want the agent to execute, while the action description provides enough information to let LLMs know when to choose the corresponding actions, as shown in Tab. 1.

We extract the set of structured actions by leveraging the powerful reasoning capability of LLMs. Specifically, a pre-trained LLM is utilized to decompose the 3141 predefined tasks provided by Mine-Dojo (Fan et al., 2022) into action sequences. Instructions for guiding LLMs on action decomposition are provided in Appendix. Then, we use their text embeddings to cluster action sequences. Finally, we extract the structured actions by selecting frequent action clusters and merging action clusters with similar functionalities. See Appendix for the set of structured actions.

## 3.2 LLM-BASED HIERARCHICAL AGENT

As illustrated in Fig. 2, the LLM-based Hierarchical Agent comprises three components: an LLM Decomposer, an LLM Planner, and an LLM Interface. These components collaborate to progressively break down the task goal into sub-goals, structured actions, and keyboard/mouse operations.

### 3.2.1 LLM DECOMPOSER

Rather than directly assigning the task goal to the agent and expecting a comprehensive and robust action plan, this work suggests the more practical strategy of decomposing the task goal into a series of more achievable sub-goals. By addressing each constituent sub-goal, the task goal can be progressively achieved. To this end, an LLM Decomposer is proposed. Goals are fed to the decomposer and recursively decomposed into a sub-goal tree. Text-based knowledge provides the necessary information for decomposition.

**Goal Format.** Since we aim to unlock the entire Overworld technology tree in Minecraft, all goals can be defined as obtaining items in the format of

$$(Object, Count, Material, Tool, Info), \qquad (2)$$

where "Object" denotes the target item, "Count" specifies the target quantity. "Material" and "Tool" refer to prerequisites needed to obtain the target item. "Info" stores the text-based knowledge related to this goal. Given a specific goal, its sentence embedding extracted from a pre-trained LLM is used to retrieve the most relevant text-based knowledge from an external knowledge base. Then, the LLM identifies the required material, tools, and related information from the gathered knowledge. The complete instructions for the LLM are described in Appendix.

**Recursive Decomposition.** This goal format enables the recursive decomposition of each goal into a sub-goal tree. Specifically, given a goal, all prerequisite items are listed as sub-goals, including materials, tools, and their corresponding quantities. Then, the recursive decomposition continues for each sub-goal until it has no prerequisites. After the decomposition, the execution sequence of the sub-goals is planned through post-order traversal. Such goal decomposition enhances the success rate of LLM planning, especially for goals necessitating long-horizon planning.

### 3.2.2 LLM PLANNER

LLMs excel at language understanding and reasoning but struggle with low-level control and multimodal perception. To leverage LLMs' strengths while addressing their limitations, we develop structured actions and feedback mechanisms as an abstract interface for them to manage agent-environment interaction. We propose an LLM-based Planner to achieve goals in Minecraft. Given a goal, it generates structured actions to control agents, receives feedback, and revises plans accordingly. It also has a text memory that aids planning by providing solutions for frequent goals.

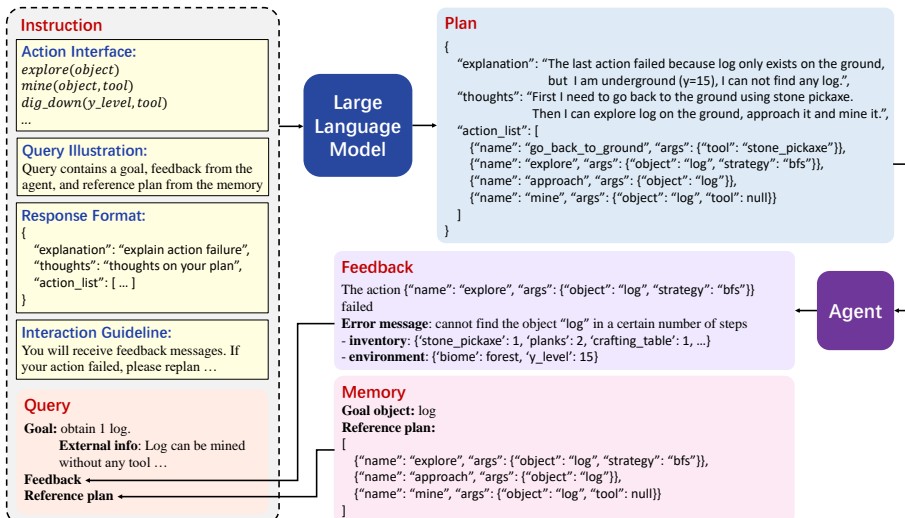

Figure 3: **Illustration of our planning process with the LLM Planner and the agent in the loop.** Given a specific goal, the planner generates plans with structured actions under the guidance of instruction, user query, previous feedback, and reference plan from memory. The agent executes the actions and provides feedback for the following planning.

**Feedback Mechanism.** Open-loop planning cannot guarantee success, especially in open-world environments, where agents might encounter unexpected events. Feedback is crucial to form an effective closed loop. Without appropriate feedback, the LLM has no information about the consequences of actions and may repeat failed action plans. Feedback message is designed to present the agent's current state in the environment (*i.e.*, inventory and environment), as well as the success and failure information for each executed action, as shown in Fig. 3. By incorporating this feedback, the LLM can update its understanding of the environment, refine strategies, and adapt its behavior accordingly.

**Planning.** Once the abstract interface is prepared, a pre-trained LLM is queried to generate a goal-specific action sequence. This is achieved through carefully designed instructions and user queries, enabling the LLM to efficiently create and revise the plans. Fig. 3 illustrates the planning process. See Appendix for the full description.

`Instruction` specifies the guidelines that LLMs must follow, including 1) *Action Interface* provides functional descriptions of the structured actions and their parameters; 2) *Query Illustration* clarifies the structure and meaning of user queries; 3) *Response Format* requires LLM to return responses in the format of {Explanation, Thought, Action List}, where "Explanation" requires LLMs to explain the reason for action failure, "Thought" requires LLM to use natural language to plan before outputting action sequences as a chain-of-thought (CoT) mechanism (Wei et al., 2022), and "Action List" outputs a list of structured actions to be executed; 4) *Interaction Guideline* guides LLMs to correct failed actions based on the feedback message, thus enabling the LLM to revise the plan.

`User Query` provides the specific query for a given goal, including 1) *Goal* represents the objective by text as "Obtain `Count Item`, given `Material` and `Tool`. Extra info: `Info`" according to Eq. equation 2; 2) *Feedback* is the feedback information of the abstract interface; 3) *Reference Plan* provides a common reference plan for the current goal retrieved from the text-base memory.

### 3.2.3 LLM INTERFACE

Unlike the existing RL-based agents that directly control the keyboard and mouse, LLM-based agents interact with the environment through structured actions and feedback messages. The LLM interface serves to implement structured actions as keyboard/mouse operations and extract observations provided by the environment into feedback messages.

Structured actions can be implemented in various ways such as hand-written scripts or RL-learned models. While RL-learned models have been employed in Minecraft previously, they were either broad in functionality but ineffective in practice or too specific in functionality, limiting their applicability to general tasks and actions. Clarifying the capability boundary of RL-learned models is

challenging. Instead, in this work, we choose to implement structured actions using hand-written scripts. Since structured actions are well-defined and easy to implement, we can manually implement them based on observations (*e.g.*, location, LiDAR, and voxel) and basic operations (*e.g.*, move, jump, adjust camera angle, click left mouse button, and click right mouse button) provided by the MineDojo (Fan et al., 2022) environment. See Appendix for details.

Feedback messages include whether the structured action execution succeeded or failed. Failure messages include bounds checks before the action execution and run-time failures after the action execution. Here we only consider the two directly known run-time failures, death and timeout. If the execution fails, the failure messages are additionally notified. Feedback also includes the current state of the agent in the environment, including the items in the inventory, the current biome, and depth, etc. See Appendix for details.

### 3.3 TEXT-BASED IN-CONTEXT LEARNING

Our agent employs LLM as its core component, potentially encountering the issue of hallucination. To improve the LLM's precision in managing physical interactions and planning, we introduce a text-based in-context learning strategy. This strategy explicitly integrates external knowledge and stores successful experiences through textual representations. Text-based knowledge will serve as context for the LLM Decomposer, while text-based memory will support the LLM Planner.

**Text-based Knowledge** is essential for automatic goal decomposition. We build an external knowledge base documented in text from the Minecraft Wiki on the Internet [1] and the item crafting/smelting recipes, providing an exhaustive source of knowledge about the Minecraft world. For instance, if we need to craft a wooden pickaxe, the crafting recipe will indicate that the required materials are three planks and two sticks and the tool is a crafting table. It also provides information about the distribution of raw materials. For example, diamonds are frequently found in levels 10∼12 underground.

**Text-based Memory** is designed for LLM to maintain common reference plans for each encountered objective as experiential knowledge. LLMs acquire experience in controlling agents and resolving specific situations through game-play and agent interaction. Instead of starting from scratch every time, using prior experience allows LLMs to handle tasks more efficiently, a process similar to human skill improvement through practice.

To this end, we design a text-based memory mechanism for LLM to store and retrieve gained knowledge. Unlike the model-driven methods, which store knowledge in parameters, this textual knowledge memory is explicit, logical, and closely aligned with human thought processes. This allows for direct application to a wide range of similar tasks, leading to more efficient learning and improved generalization.

**Text-based In-context Learning** is designed to improve the LLM Planner. Specifically, during each game episode, once the goal is achieved, the entirely executed action list would be stored in memory. The LLM may achieve the same goal under various circumstances, resulting in a range of different plans. To identify a common reference plan suitable for general situations, essential actions from multiple plans are summarized. This summarization process is also implemented using LLMs (see Appendix for details). When encountering similar goals, the LLM creates new plans based on the summarized reference plans retrieved from memory. Successful action sequences from these new plans are also added to memory for future summarization. As the LLM-based Planner accumulates summaries, it becomes increasingly effective.

## 4 EXPERIMENTS

**Task Definition and Metrics.** We measure the ability of GITM through item collection tasks. Collecting all in-game items clearly demonstrates the adaptability of our agent, as it requires interaction with diverse terrains, biomes, and mobs. We only collect items could be found in the Overworld. We exclude items could only be obtained by trading with villagers, opening treasure chest or find a special structure on the map. This give us a total of 262 tasks. For the assessment of our agent, we employ "Coverage of the Overworld Technology Tree" and"Success Rate" as evaluation metrics.

---

[1] https://minecraft-archive.fandom.com/wiki/Minecraft_Wiki

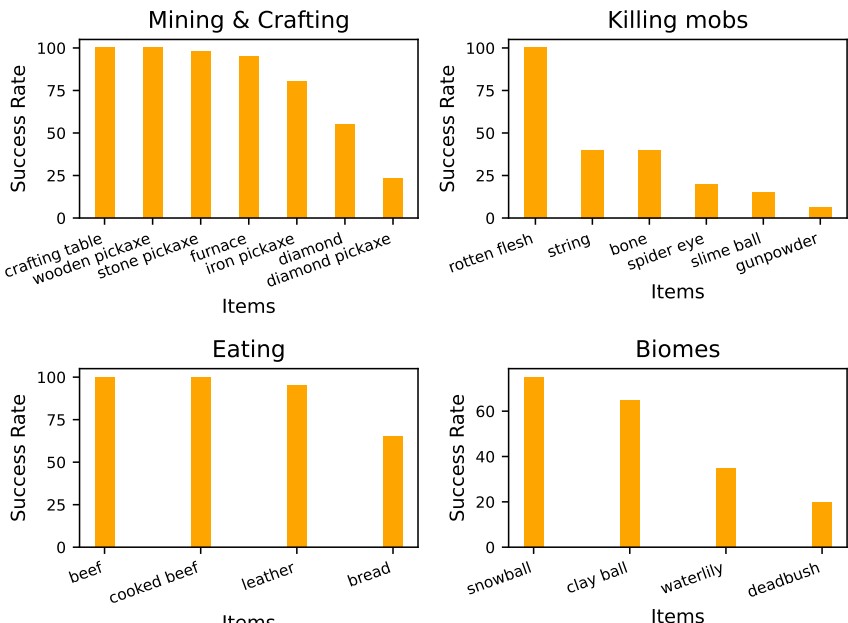

Figure 4: **Success rate for typical items in the entire Minecraft Overworld Technology Tree**. Our agent is able to handle a wide range of tasks that are required to effectively navigate the open-world, including mining, crafting, killing mobs, eating and exploring biomes.

**Implementation Details.** In this paper, we take the classic robotics approach (Brooks, 1986) of dividing an agent into three parts: sensing, planning, and acting. We focus on using large language models (LLMs) to handle the planning challenges in Minecraft. Rather than work on perception and control, we use more reliable off-the-shelf tools for those parts (lidar scans provided by MineDojo (Fan et al., 2022) for sensing and pre-programmed actions for control). This lets us concentrate on using LLMs for planning without getting bogged down by limitations in perception and control. By relying on oracle for sensing and acting, we can better evaluate how well LLMs perform at planning tasks in the Minecraft environment. Overall, our goal is to show how capable LLMs are at planning by focusing on that while using simple but effective solutions for perception and control.

### 4.1 MAIN RESULT

**Unlocking the Overworld Technology Tree.** Compared with existing Minecraft agents (Baker et al., 2022; Hafner et al., 2023; Wang et al., 2023) which mostly focuses on solving the `ObtainDiamond` task and could only unlock a limited part of the full technology tree (13/262 for Dreamerv3, 15/262 VPT, 69/262 for DEPS), our approach could collect 99.2% items of the technology tree as shown in Fig. 1. Previous methods have weak abilities to generalize to unseen tasks and solve extremely long-horizon tasks (e.g., obtaining a "enchanted_book"). To deal with these challenges, we extract a well-defined set of structured actions by using LLMs to decompose over 3141 predefined MineDojo tasks. This provides broad, open-world Minecraft capability. Combined with LLM planning, it enables solving more complex tasks than `ObtainDiamond`. We note that crafting TNT items appears to be still challenging for our agent, because it requires to kill multiple creepers. Strengthening the combat capability is a future direction for more powerful agents.

**Success Rate for the Technology Tree.** We show the success rate of our method for collecting typical items in Fig. 4. To effectively navigate the open-world, the agent must learn to gather resources, craft tools, fight with mobs, eat to recover health and explore different biomes. Our method can achieve decent success rate for most of these tasks, enabling the agent to adapt to the ever-changing environment. We provide the success rate for all Overworld items in the appendix.

### 4.2 COMPARISON WITH OTHER MINECRAFT AGENTS

In Tab. 2, we test different agents on the well known `ObtainDiamond` challenge, i.e, obtaining a diamond from scratch in Minecraft. Based on the abstract interface provided by our designed structured action, we implement some representative LLM-based agents including ReAct (Yao et al.,

Table 2: Comparison of our GITM with previous methods on the `ObtainDiamond` challenge. The milestone items from left to right are crafting table 🧰, wooden pickaxe ⛏, stone pickaxe ⛏, iron pickaxe ⛏, and diamond 💎. Note that the LLM-based methods cannot be directly compared with RL-based methods (with grey text) since they have different observations and actions. [†] Both AutoGPT and our GITM have a memory mechanism. The "-zero" agents are tested with empty memory initialized in each episode.

| Method | Observation | Action | Learning steps | Success Rate (%) 🧰 | ⛏ | ⛏ | ⛏ | 💎 |
|--------|-------------|--------|----------------|------|------|------|------|------|
| DreamerV3 | RGB, status | low-level | ∼1e8 | - | 50.0 | 3.0 | 0.01 | 0.01 |
| DEPS | RGB, voxels, status | low-level | - | 90.0 | 80.0 | 73.3 | 10.0 | 0.6 |
| VPT | RGB | low-level | ∼1e10 | 100.0 | 100.0 | 100.0 | 85.0 | 20.0 |
| ReAct | LiDAR, voxels, status | structured | 0 | 30.0 | 15.0 | 0.0 | 0.0 | 0.0 |
| AutoGPT-zero[†] | LiDAR, voxels, status | structured | 0 | 35.0 | 17.5 | 2.5 | 0.0 | 0.0 |
| AutoGPT | LiDAR, voxels, status | structured | ∼1e5 | 50.0 | 30.0 | 15.0 | 0.0 | 0.0 |
| **GITM**-zero[†] | LiDAR, voxels, status | structured | 0 | 100.0 | 100.0 | 82.5 | 62.5 | 32.5 |
| **GITM** | LiDAR, voxels, status | structured | ∼1e5 | 100.0 | 100.0 | 97.5 | 80.0 | 55.0 |

2022) and AutoGPT (Significant-Gravitas/AutoGPT:, 2023) and compare them with our GITM. We also report the results of existing RL-based agents as reference, including VPT (Baker et al., 2022), DreamerV3 (Hafner et al., 2023), and DEPS (Wang et al., 2023). Note that we do not directly compare the success rate of our method with RL-based methods since their observations and actions are different. Detailed observation and action spaces of different methods are listed in Appendix.

ReAct (Yao et al., 2022) is a common approach for LLM-based agent which synthesizes reasoning and action with chain-of-thought prompting (Wei et al., 2022). It can only handle simple tasks such as obtaining crafting table and wooden pickaxe with low success rates. This indicates that even though the physical interaction with the environment is mainly handled by the structured actions, planning is still challenging in Minecraft. The difficulty of planning arises from the long-horizon complex tasks, as well as the unpredictable nature of the open-world environment including unexpected events and imperfections of the structured actions in perception and control. AutoGPT (Significant-Gravitas/AutoGPT:, 2023) decomposes tasks into high-level goals and short-term plans and executes actions in a ReAct-style loop. Utilizing high-level plans, it unlocks more complex tasks such as obtaining stone pickaxe. Besides, AutoGPT leverages memory mechanism to store and retrieve useful information in previous experience. The success rates increase with a learned memory (*e.g.*, from 2.5% to 15.0% for stone pickaxe).

Our GITM is the only LLM-based agent that successfully obtains diamond and iron pickaxe with success rates 80.0% and 55.0%, respectively. On simple tasks, GITM achieves success rates equal or approximate to 100%, greatly outperforming existing LLM-based methods. Furthermore, GITM gains significant boosts with the learned memory (+17.5% for iron pickaxe and +22.5% for diamond in success rate), confirming the effectiveness of our proposed text-based in-context learning. Our method is superior because it recursively decomposes goals into sub-goals, enabling the agent to learn from simple to complex tasks. The experience gained on easier sub-goals directly informs solutions for harder goals. Additionally, our agent summarizes and refines general plans in memory, rather than simply recording all history which may include distracting errors and hallucinations.

Existing RL-based agents have also achieved considerable success rates on this task. For example, VPT (Baker et al., 2022) gets 85.0% for iron pickaxe and 20.0% for diamond. However, it takes tens of millions of steps to for an RL agent to converges to meaningful non-zero success rates, and the learned agents can not be easily re-tasked for other tasks.

### 4.3 ABLATION STUDY

We conduct ablation experiments on the `ObtainDiamond` task. When leveraging goal decomposition, for each sub-goal, we limit the maximum number of queries to LLM as 30, and exceeding the query limit will be counted as a failure. For each setting, we run 40 games and calculate the success rate. We report the success rates of achieving the milestone items, including crafting table, wooden pickaxe, stone pickaxe, iron pickaxe, and diamond.

Table 3: **Ablation study of structured action.** "w/o structured action" means having the LLM planner directly output low-level controls instead of structured actions. "w/o feedback message" indicates that the output from executing structured actions is not reported back to the LLM planner. "RL w/ structured action" refers to training an agent with RL on structured actions.

| | Setting | Success Rate (%) | | | | |
|---|---|---|---|---|---|---|
| | | 🟫 | ↗ | ↗ | ↗ | 💎 |
| (a) | w/o structured action | 0.0 | 0.0 | 0.0 | 0.0 | 0.0 |
| (b) | w/o feedback message | 100.0 | 97.5 | 85.0 | 12.5 | 2.5 |
| (c) | GITM | 100.0 | 100.0 | 97.5 | 80.0 | 55.0 |
| (d) | RL w/ structured action | 0.0 | 0.0 | 0.0 | 0.0 | 0.0 |

Table 4: **Ablations of LLM decomposer, text-based external knowledge, and text-based memory.**

| | Setting | Success Rate (%) | | | | |
|---|---|---|---|---|---|---|
| | | 🟫 | ↗ | ↗ | ↗ | 💎 |
| (a) | w/o decomposer | 100.0 | 97.5 | 50.0 | 0.0 | 0.0 |
| (b) | w/o external knowledge | 100.0 | 100.0 | 82.5 | 55.0 | 30.0 |
| (c) | w/o memory | 100.0 | 100.0 | 80.0 | 60.0 | 32.5 |
| (d) | GITM | 100.0 | 100.0 | 97.5 | 80.0 | 55.0 |

**Structured Action.** Tab. 3 studies the importance of structured action. Tab. 3(a) and (c) shows that, without structured action, LLM can not produce reasonable actions due to the lack of environment grounding. Feedback contains the agent's state and the execution result of the actions, which helps the planner to understand and make another attempt to correct the mistakes in the previous and deal with corner cases. This enables the planner to accomplish a broader range of goals with higher success rate. As shown in Tab. 3(b) and (c), our agent enhances the ability to collect diamond by combining feedback with structured action. We also make an initial attempt to apply reinforcement learning on structured actions in Tab. 3(d). We are not able to obtain a reasonable result with similar learning steps of our method. We suspect the structured action space is too large for RL methods to learn from scratch (note the structured action parameters should also be considered).

**Goal Decomposition.** Without goal decomposition, the planner achieved only short-term tasks like obtaining stone axes, and only at a 50% success rate (Tab. 4(a)). This demonstrates the necessity of goal decomposition. Leveraging the powerful long-term planning capabilities of LLMs, the goals are decomposed into sub-goals feasible and practical for the planner, so the agent is able to obtain diamond with a 55% success rate.

**External Knowledge Base.** External knowledge contains general rules, crafting recipes, and common tricks in Minecraft, such as the recipes for crafting iron ingot and iron pickaxe, the suitable location to find diamond ore, and the efficient way to get cobblestone. Providing the planner with this information greatly boosts the success rate of obtaining iron pickaxe and diamond, and the success rate of mining diamond increase by 25% by learning from the knowledge base that diamonds are more likely to appear in specific levels (Tab. 4(b)).

**Text-based Memory.** Leveraging the reference plans in the memory, the planner handles the tasks it has encountered more efficiently. The success rates of obtaining iron pickaxe and diamond are 80.0% and 55.0%, surpassing the model without memory by 20.0% and 22.5%, respectively (Tab. 4(c)).

## 5  CONCLUSION

We introduce the GITM framework for Minecraft, which utilizes Large Language Models (LLMs) with text-based knowledge and memory for hierarchical planning of goals. A set of structured actions is constructed to enable LLMs to interact within Minecraft using textual descriptions. LLM Decomposer, LLM Planner, and LLM Interface are introduced to gradually decompose goals into sub-goals, structured actions, and keyboard/mouse operations. A text-based in-context learning is designed to enhance future planning. By achieving decent performance and impressive learning efficiency, this work makes significant progress towards adaptive agents that can effectively navigate, make decisions, and interact in the open-world and ever-evolving environment of Minecraft.

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

# A IMPLEMENTATION DETAILS

## A.1 ACTION ABSTRACTION

**Instruction for Extracting Structured Actions.** To extract structured actions, we first ask LLM to generate a tree-structured action planning for each of the 3141 predefined tasks provided by MineDojo, and then converts each action step into a (`verb`, `object`, `tool`, `material`) tuple. During decomposition, it is essential to ensure actions are neither too broad nor too specific. We adjusted the depth of the action decomposition tree to achieve balance, and empirically set the depth as 2 to meet our requirements.

Specifically, we use `gpt-3.5-turbo` from OpenAI API to generate the structured actions. We add the following instruction to the content of "SYSTEM" role to generate the tree-structured plan. We add the goal description, *e.g.*, "find material and craft a iron pickaxe", to the content of "USER" role and then asks LLM to response according to the requirements.

---

**SYSTEM:**
You serve as an assistant that helps me play Minecraft.

I will give you my goal in the game, please break it down as a tree-structure plan to achieve this goal.

The requirements of the tree-structure plan are:

1. The plan tree should be exactly of depth 2.
2. Describe each step in one line.
3. You should index the two levels like '1.', '1.1.', '1.2.', '2.', '2.1.', etc.
4. The sub-goals at the bottom level should be basic actions so that I can easily execute them in the game.

**USER:**
The goal is to {`goal description`}. Generate the plan according to the requirements.

---

After that, we extract the action tuple from each sentence of the leaf nodes. We use the following instruction as the content of "SYSTEM" role to extract the tuple and add the sentence to the content of "USER" role.

---

**SYSTEM:**
You serve as an assistant that helps me play Minecraft.

I will give you a sentence. Please convert this sentence into one or several actions according to the following instructions.

Each action should be a tuple of four items, written in the form ('verb', 'object', 'tools', 'materials')

'verb' is the verb of this action.
'object' refers to the target object of the action.
'tools' specifies the tools required for the action.
'material' specifies the materials required for the action.
If some of the items are not required, set them to be 'None'.

**USER:**
The sentence is {`sentence`}. Generate the action tuple according to the requirements.

---

Then, we extract the structured actions by selecting frequent actions and merging actions with similar functionalities. The set of structured actions is {`equip`, `explore`, `approach`, `mine/attack`, `dig_down`, `go_up`, `build`, `craft/smelt`, `apply`}. Note that we disregard more detailed action decomposition for `attack` and `build` to remove overly detailed short-term actions and focus on long-term task completion.

## A.2 LLM-BASED HIERARCHICAL AGENT

### A.2.1 LLM DECOMPOSER

We use `gpt-3.5-turbo` from OpenAI API [2] for goal decomposition. The prompt is shown as follows, which consists of two parts: instruction with the role of "SYSTEM" and query with the role of "USER". The {`object quantity`}, {`object name`} and {`related knowledge`} are injectable slots that will be replaced with corresponding texts before fed into the LLM.

---

**SYSTEM:**

You are an assistant for the game Minecraft.

I will give you some target objects and some knowledge related to the object. Please write the obtaining of the object as a goal in the standard form.

The standard form of the goal is as follows:
{
  "object": "the name of the target object",
  "count": "the target quantity",
  "material": "the materials required for this goal, a dictionary in the form {material_name: material_quantity}. If no material is required, set it to None",
  "tool": "the tool used for this goal. If multiple tools can be used for this goal, only write the most basic one. If no tool is required, set it to None",
  "info": "the knowledge related to this goal"
}
The information I will give you:
Target object: the name and the quantity of the target object
Knowledge: some knowledge related to the object.

Requirements:
1. You must generate the goal based on the provided knowledge instead of purely depending on your own knowledge.
2. The "info" should be as compact as possible, at most 3 sentences. The knowledge I give you may be raw texts from Wiki documents. Please extract and summarize important information instead of directly copying all the texts.

Goal Example:
{
  "object": "iron_ore",
  "count": 1,
  "material": None,
  "tool": "stone_pickaxe",
  "info": "iron ore is obtained by mining iron ore. iron ore is most found in level 53. iron ore can only be mined with a stone pickaxe or better; using a wooden or gold pickaxe will yield nothing."
}
{
  "object": "wooden_pickaxe",
  "count": 1,
  "material": {"planks": 3, "stick": 2},
  "tool": "crafting_table",
  "info": "wooden pickaxe can be crafted with 3 planks and 2 stick as the material and crafting table as the tool."
}

**USER:**

---

[2]https://platform.openai.com/docs/api-reference

> Target object: {`object quantity`} {`object name`}
> Knowledge: {`related knowledge`}

The recursive decomposition generates a sub-goal tree starting from the final goal object as the root node: if a goal has some prerequisites (materials or tools), for each required material or tool, we add a child node representing the goal of obtaining that material or tool, and then recursively decompose the child node, until there is no more prerequisites. The related knowledge is from: 1) Crafting/smelting recipes in MineDojo (Fan et al., 2022), written in the form "Crafting {`quantity`} {`object`} requires {`material`} as the material and {`tool`} as the tool"; 2) Wiki on the Internet [3]. We extract the paragraphs with keywords "obtaining", "mining", "sources", etc.

### A.3   LLM PLANNER

Here we present the prompt for planning with LLM. We also use `gpt-3.5-turbo` from OpenAI API as the LLM planner. The model accepts inputs in the form of a chat, i.e., the prompt is a dialogue consisting of several messages, each of which contains a role and the content. We set the `Instruction` with the role "SYSTEM" at the beginning, and use the `User Query` with the role "USER" to query the LLM for response. The content of the `Instruction` and `User Query` are as follows.

#### A.3.1   INSTRUCTION

> **SYSTEM:**
>
> You serve as an assistant that helps me play the game Minecraft.
>
> I will give you a goal in the game. Please think of a plan to achieve the goal, and then write a sequence of actions to realize the plan. The requirements and instructions are as follows:
>
> 1. You can only use the following functions. Don't make plans purely based on your experience, think about how to use these functions.
>
> `explore(object, strategy)`
> Move around to find the object with the strategy: used to find objects including block items and entities. This action is finished once the object is visible (maybe at a distance).
> Augments:
> - object: a string, the object to explore.
> - strategy: a string, the strategy for exploration.
>
> `approach(object)`
> Move close to a visible object: used to approach the object you want to attack or mine. It may fail if the target object is not accessible.
> Augments:
> - object: a string, the object to approach.
>
> `craft(object, materials, tool)`
> Craft the object with the materials and tool: used for crafting new object that is not in the inventory or is not enough. The required materials must be in the inventory and will be consumed, and the newly crafted objects will be added to the inventory. The tools like the crafting table and furnace should be in the inventory and this action will directly use them. Don't try to place or approach the crafting table or furnace, you will get failed since this action does not support using tools placed on the ground. You don't need to collect the items after crafting. If the quantity you require is more than a unit, this action will craft the objects one unit by one unit. If the materials run out halfway through, this action will stop, and you will only get part of the objects you want that have been crafted.
> Augments:
> - object: a dict, whose key is the name of the object and value is the object quantity.

---

[3]`https://minecraft-archive.fandom.com/wiki/Minecraft_Wiki`

- materials: a dict, whose keys are the names of the materials and values are the quantities.
- tool: a string, the tool used for crafting. Set to null if no tool is required.

`mine(object, tool)`
Mine the object with the tool: can only mine the object within reach, cannot mine object from a distance. If there are enough objects within reach, this action will mine as many as you specify. The obtained objects will be added to the inventory.
Augments:
- object: a string, the object to mine.
- tool: a string, the tool used for mining. Set to null if no tool is required.

`attack(object, tool)`
Attack the object with the tool: used to attack the object within reach. This action will keep track of and attack the object until it is killed.
Augments:
- object: a string, the object to attack.
- tool: a string, the tool used for mining. Set to null if no tool is required.

`equip(object)`
Equip the object from the inventory: used to equip equipment, including tools, weapons, and armor. The object must be in the inventory and belong to the items for equipping.
Augments:
- object: a string, the object to equip.

`digdown(object, tool)`
Dig down to the y-level with the tool: the only action you can take if you want to go underground for mining some ore.
Augments:
- object: an int, the y-level (absolute y coordinate) to dig to.
- tool: a string, the tool used for digging. Set to null if no tool is required.

`go_back_to_ground(tool)`
Go back to the ground from underground: the only action you can take for going back to the ground if you are underground.
Augments:
- tool: a string, the tool used for digging. Set to null if no tool is required.

`apply(object, tool)`
Apply the tool on the object: used for fetching water, milk, lava with the tool bucket, pooling water or lava to the object with the tool water bucket or lava bucket, shearing sheep with the tool shears, blocking attacks with the tool shield.
Augments:
- object: a string, the object to apply to.
- tool: a string, the tool used to apply.

2. You cannot define any new function. Note that the "Generated structures" world creation option is turned off.

3. There is an inventory that stores all the objects I have. It is not an entity, but objects can be added to it or retrieved from it anytime at anywhere without specific actions. The mined or crafted objects will be added to this inventory, and the materials and tools to use are also from this inventory. Objects in the inventory can be directly used. Don't write the code to obtain them. If you plan to use some object not in the inventory, you should first plan to obtain it. You can view the inventory as one of my states, and it is written in form of a dictionary whose keys are the name of the objects I have and the values are their quantities.

4. You will get the following information about my current state:
- inventory: a dict representing the inventory mentioned above, whose keys are the name of the objects and the values are their quantities
- environment: a string including my surrounding biome, the y-level of my current location, and whether I am on the ground or underground

Pay attention to this information. Choose the easiest way to achieve the goal conditioned on my current state. Do not provide options, always make the final decision.

5. You must describe your thoughts on the plan in natural language at the beginning. After that, you should write all the actions together. The response should follow the format:
{
    "explanation": "explain why the last action failed, set to null for the first planning",
    "thoughts": "Your thoughts on the plan in natural languag",
    "action_list": [
        {"name": "action name", "args": {"arg name": value}, "expectation": "describe the expected results of this action"},
        {"name": "action name", "args": {"arg name": value}, "expectation": "describe the expected results of this action"},
        {"name": "action name", "args": {"arg name": value}, "expectation": "describe the expected results of this action"}
    ]
}
The action_list can contain arbitrary number of actions. The args of each action should correspond to the type mentioned in the Arguments part. Remember to add "'dict'" at the beginning and the end of the dict. Ensure that you response can be parsed by Python json.loads

6. I will execute your code step by step and give you feedback. If some action fails, I will stop at that action and will not execute its following actions. The feedback will include error messages about the failed action. At that time, you should replan and write the new code just starting from that failed action.

### A.3.2   USER QUERY

**USER:**

My current state:
- inventory: {inventory}
- environment: {environment}

The goal is to {goal}.

Here is one plan to achieve similar goal for reference: {reference plan}.

Begin your plan. Remember to follow the response format.
*or* Action {successful action} succeeded, and {feedback message}. Continue your plan. Do not repeat successful action. Remember to follow the response format.
*or* Action {failed action} failed, because {feedback message}. Revise your plan from the failed action. Remember to follow the response format.

### A.3.3   LLM INTERFACE

**Action Implementation.** The observation of the action contains LiDAR rays with an interval of 5 degrees in the horizon and vertical direction for locating objects, and voxels with 10 unit radius only for navigation, inventory, life status, and agent location status (X-ray cheating is carefully avoided). RGB is not used in our implementation, although it provides more information than LiDAR rays. For example, the biome, and category of the dropping item can not be identified by LiDAR rays. Some objects may also be missed by LiDAR due to the sparseness of LiDAR rays. Different from Hafner et al. (2023) who set the breaking speed to 100, we did not change the game settings. The detailed implementation of each structured action is as follows:

- `equip`: The equip action calls the environment API to equip the required object. The action succeeds when the API returns success. The action fails when the object is not in inventory or the equip API returns failure.
- `explore`: The explore action traverses the world until the object is visible. This action regards the world as a chessboard, and each node on the chessboard is the center point of a $20 \times 20$ units

area. Two strategies are implemented depending on whether the agent is on the ground or not. When the agent is on the ground, the BFS explore will be adopted. When the agent is under the ground, mainly for exploring ore, the DFS explore will be adopted. In the DFS exploration, the agent will break the blocks to form a mine road with a width of 1 and a height of 2. The action succeeds when the object is visible. The action fails when the explore exceeds a preset steps of 10,000 but the required object is not found.

- `approach`: The approach action finds the nearest visible required object and walks towards the object. We adopt $A^*$ algorithm for finding a path. The $A^*$ algorithm can jump, translate, and fall in four directions of north, south, east and west. We also allow the agent to jump while placing a block under the agent for ascent. If the object is out of the voxel observation range, $A^*$ algorithm is iteratively applied to find the location nearest to the object. The action succeeds when the $\ell^\infty$ norm distance between the object and agent is less than 2. The action fails when there is no required object visible or no path can be found to walk close to the object.

- `mine/attack`: The mine/attack action uses the keyboard attack API with the tools to attack the object. Only visible objects could be mined or attacked. The object of mine should be blocks, and the agent will continue mining the block until it is broken. The object of attack should be entities, and the agent will iteratively approach and attack the entity until it is killed. After the block is broken or the entity is killed, if there are items dropped by them, the agent will approach the items to collect them. The action succeeds when the block is broken or the entity is killed. The action fails when there is no visible object, no required tools is in inventory, or the visible object is out of attack range.

- `dig_down`: The dig_down action iteratively breaks the block underfoot with the tool until the required ylevel is reached. If the agent is on the ground, before digging down, the current location is stored for going up action. After the action succeeds, the state of the agent is set to underground. The action succeeds when the required ylevel is reached. The action fails when it exceeds the reset max steps 10,000 or no required tool is in inventory.

- `go_up`: The agent will first go back to the location stored by dig_down. Then, the go_up action puts dirt blocks underfoot to raise the agent. After the action is finished, the state of agent is set to on the ground. The action succeeds when the pre-stored location is reached. The action fails when the walk fails, exceeds the reset max steps 10,000 or there is no required tool in inventory.

- `build`: The build action places the required blocks according to a given blueprint from bottom to up. The action succeeds when all blocks have been placed. The action fails when there are no enough materials in inventory or it is invalid to place some blocks.

- `craft/smelt`: The action calls the environment API to craft/smelt the required object. The action succeeds when the required object is obtained. The actions fail when there are no enough materials in inventory or the agent is unable to place the crafting table/furnace or the API fails.

- `apply`: The apply action calls the keyboard use API, and applies the specific tool to the object, *e.g.*, applying the bucket on water to obtain water bucket. The action succeeds when the API returns success. The action fails when there is no visible object, no tool in inventory or the API fails.

**Feedback Message.** After the execution of each action, we will get feedback from the structured actions. The feedback will refresh the agent's state in Sec. A.3.2, including current inventory, biome, ylevel, and on/under the ground status. The feedback will also contain the success/fail message from these actions, as well as the inventory change during the action.

## A.4 MEMORY

### A.4.1 LEARNING PROCESS

We maintain the text-based memory with a dictionary, whose keys are sub-goals and values are lists of successful action sequences for the corresponding sub-goals. The construction and update of the memory are through the following learning process:

- When encountering a new sub-goal that is not in the memory, the LLM planner creates plans without reference. Once the sub-goal is achieved, the entirely executed action sequence will be stored in the memory.

- When encountering a sub-goal with memory, the first action sequence in the recording list for this goal is retrieved as the reference plan, with which the LLM planner tries to achieve the goal. If it succeeds, the newly executed action sequence will be added to the last of the recording list.

- For each sub-goal, once the number of action sequences recorded in its list reaches $N$, we pop all the $N$ sequences and use LLM to summarize them into a common plan solution suitable for various scenarios, which is then put first in the list. $N$ is set to 5 in all our experiments.

To learn the memory for obtaining all items, starting from scratch each time would take a long time. In addition, it is necessary to avoid spending most of the time on learning simple tasks and not investing enough in learning difficult tasks. To improve learning efficiency, we suggest studying the sub-goals individually one by one. We first use our LLM Decomposer to generate sub-goal trees for all items, acquiring the set of all sub-goals involved. Then for each sub-goal, the LLM planner plays multiple times given its prerequisites including the required materials and tools. The learning process of the sub-goal is finished once we obtain $N = 5$ successful action sequences and summarize them into one common plan solution for reference.

### A.4.2 IMPLEMENTATION OF MEMORY SUMMARIZATION

We also use `gpt-3.5-turbo` from OpenAI API for memory summarization but in a different dialogue. We use the following prompt to instruct the summarization with the role "SYSTEM". The slot {`action description`} is replaced with the same descriptions of interfaces of the structured actions as Sec. A.3.1. We list all the action sequences to be summarized in the query with the role "USER", which is fed into the LLM for response.

---

**SYSTEM:**

You serve as an assistant that helps me play the game Minecraft.

I am using a set of actions to achieve goals in the game Minecraft. I have recorded several action sequences successfully achieving a goal in a certain state. I will give you the goal, the state, and the sequences later. Please summarize the multiple action sequences into a single action sequence as a universal reference to achieve the goal given that certain state. Here are the instructions:

1. Each action sequence is a sequence of the following actions:

{`action description`}

2. The action sequences before and after summarization are always conditioned on the given state, i.e., the actions are taken in that certain state to achieve the goal. I will describe the state in the following form: State: - inventory: a dict whose keys are the name of the objects and the values are their quantities. This inventory stores all the objects I have. - environment: a dict including my surrounding biome and whether I am on the ground or underground.

3. The action sequence you summarize should be able to achieve the goal in general cases without specific modification. Every necessary action should be included, even though it does not appear in some sequences because I manually skipped it in some lucky cases. The actions redundant or irrelevant to the goal should be filtered out. The corner cases, such as success by luck and dealing with contingencies, should not be summarized into the final sequence.

4. You should describe your thoughts on summarization in natural language at the beginning. After that, give me the summarized action sequence as a list in JSON format. Your response should follow this form:

Thoughts: "Your thoughts and descriptions of your summarization"
Summarized action sequence:
[
    {"name": "action name", "args": {"arg name": value}, "expectation": "describe the expected results of this action"},
    {"name": "action name", "args": {"arg name": value}, "expectation": "describe the expected results of this action"},
    {"name": "action name", "args": {"arg name": value}, "expectation": "describe the

---

```
expected results of this action"}
]
```

## B    OBSERVATION AND ACTION SPACES

We list the observation and action spaces of different methods in Tab. 5. Prior RL-based agents take raw images as input and use low-level controls, while our agent accepts oracle inputs and uses structured actions. We only use voxel information of the blocks on the surface without X-ray cheating.

Table 5: **Observation and output spaces of different methods.**

| Method | Perception Observation | Status Observation | Output Space |
|---|---|---|---|
| VPT | camera view RGB | | keyboard/mouse (20 keys, mouse movements) |
| DreamerV3 | camera view RGB | inventory life status | keyboard/mouse & GUI-free crafting (25 actions based on MineRL ObtainDiamond) |
| DEPS | camera view RGB block voxel (3 x 3 x 3) | yaw/pitch angle GPS location | keyboard/mouse & GUI-free crafting (42 actions discretized from MineDojo) |
| **GITM (ours)** | LiDAR rays (interval = 5") block voxel (radius = 10, without X-ray cheating) | inventory life status biome agent position | action APIs (9 APIs manually implemented on MineDojo) |

## C    RESULTS OF ALL ITEMS

We provide the success rate of all items in the entire Minecraft Overworld Technology Tree in Tab. 6.

**Experiment Setting.** Considering the large number of items, including those difficult to be obtained, we implemented an incremental testing strategy. This strategy is designed to keep the testing costs within a reasonable range, while also accounting for the rarity of certain items. We avoided a uniform increase in the number of tests across all items to accommodate the hardest-to-obtain ones, which would have resulted in prohibitive testing costs. Instead, we employed a incremental testing process.

For each item, we begin with 20 games. If the success count is less than or equal to 1, we increase to 50 games. If the success count remains less than or equal to 1, we further increase to 100, and eventually 200 games. This testing continues until the success count finally exceeds 1, or we complete 200 games. By following this efficient strategy, we ensure a cost-effective and reliable evaluation of each item, regardless of its availability. Moreover, because some items need long-term planning and crafting chain, we do not set restrictions on the time limit or query limit.

**Exploring Biome.** Biomes can be a key factor that strongly influences the success rate. Some items, like cactus, pumpkin, or melon, can only be found in specific biomes. The distribution of biomes highly limits the success rate of some items.

Table 6: **Success rate for all 262 items in the entire Minecraft Overworld Technology Tree.**

| Item Name | Success Rate | Item Name | Success Rate | Item Name | Success Rate | Item Name | Success Rate |
|---|---|---|---|---|---|---|---|
| acacia boat | 100 | stonebrick | 100 | milk bucket | 65 | cactus | 20 |
| acacia door | 100 | trapdoor | 100 | coal block | 65 | activator rail | 15 |
| acacia fence | 100 | wooden axe | 100 | gravel | 65 | detector rail | 15 |
| acacia fence gate | 100 | wooden button | 100 | water bucket | 60 | diamond helmet | 15 |
| acacia stairs | 100 | wooden door | 100 | iron bars | 60 | slime ball | 15 |
| beef | 100 | wooden hoe | 100 | iron door | 60 | gold ingot | 15 |
| birch boat | 100 | wooden pickaxe | 100 | rail | 60 | gold nugget | 15 |
| birch door | 100 | wooden pressure plate | 100 | flower pot | 60 | gold ore | 15 |
| birch fence | 100 | wooden shovel | 100 | cauldron | 60 | golden shovel | 15 |
| birch fence gate | 100 | wooden slab | 100 | iron leggings | 60 | deadbush | 15 |
| birch stairs | 100 | wooden sword | 100 | flint | 55 | red mushroom block | 15 |
| boat | 100 | armor stand | 100 | arrow | 55 | golden hoe | 15 |
| bowl | 100 | rotten flesh | 100 | iron chestplate | 55 | golden sword | 15 |
| chest | 100 | stone slab | 100 | iron block | 55 | light weighted pressure plate | 15 |
| chicken | 100 | stone slab2 | 100 | brick block | 55 | diamond leggings | 15 |
| cobblestone | 100 | red sandstone stairs | 100 | clay | 55 | pumpkin | 15 |
| cobblestone wall | 100 | sandstone stairs | 100 | hardened clay | 55 | pumpkin seeds | 15 |
| cooked beef | 100 | feather | 100 | red flower | 50 | brown mushroom block | 15 |
| cooked chicken | 100 | rabbit foot | 100 | yellow flower | 50 | mushroom stew | 10 |
| cooked mutton | 100 | item frame | 95 | egg | 50 | emerald | 10 |
| cooked porkchop | 100 | leather | 95 | hay block | 45 | lit pumpkin | 10 |
| crafting table | 100 | leather boots | 95 | flint and steel | 45 | golden axe | 10 |
| dark oak boat | 100 | leather helmet | 85 | hopper minecart | 45 | golden pickaxe | 10 |
| dark oak door | 100 | sapling | 80 | apple | 45 | golden boots | 10 |
| dark oak fence | 100 | tallgrass | 80 | beetroot | 40 | repeater | 9 |
| dark oak fence gate | 100 | wheat | 80 | beetroot seeds | 40 | carrot on a stick | 9 |
| dark oak stairs | 100 | wheat seeds | 80 | string | 40 | melon | 8 |
| dirt | 100 | iron ingot | 80 | diamond | 40 | melon seeds | 8 |
| double plant | 100 | iron nugget | 80 | diamond shovel | 40 | obsidian | 7 |
| fence | 100 | iron ore | 80 | jukebox | 40 | golden helmet | 7 |
| fence gate | 100 | iron shovel | 80 | bone | 40 | diamond chestplate | 7 |
| furnace | 100 | shield | 80 | bone meal | 40 | anvil | 7 |
| glass bottle | 100 | trapped chest | 80 | red mushroom | 35 | map | 7 |
| glass pane | 100 | tripwire hook | 80 | diamond hoe | 35 | writable book | 6 |
| jungle boat | 100 | grass | 80 | diamond sword | 35 | redstone block | 6 |
| jungle door | 100 | heavy weighted pressure plate | 80 | lava bucket | 35 | gunpowder | 6 |
| jungle fence | 100 | iron hoe | 80 | paper | 35 | bow | 6 |
| jungle fence gate | 100 | iron sword | 80 | reeds | 35 | golden carrot | 5 |
| jungle stairs | 100 | leaves | 80 | sugar | 35 | cake | 4 |
| ladder | 100 | painting | 80 | waterlily | 35 | sticky piston | 4 |
| lever | 100 | shears | 80 | baked potato | 35 | bone block | 4 |
| log | 100 | wool | 80 | potato | 35 | golden leggings | 3 |
| mutton | 100 | leather leggings | 80 | carrot | 35 | diamond block | 3 |
| oak stairs | 100 | coal | 75 | brown mushroom | 35 | clock | 3 |
| planks | 100 | torch | 75 | book | 35 | melon block | 3 |
| porkchop | 100 | snow | 75 | dropper | 30 | fermented spider eye | 2 |
| rabbit hide | 100 | snow layer | 75 | noteblock | 30 | pumpkin pie | 2 |
| red sandstone | 100 | snowball | 75 | redstone | 30 | golden rail | 2 |
| sandstone | 100 | bucket | 75 | redstone torch | 30 | fireworks | 2 |
| sign | 100 | iron axe | 75 | beetroot soup | 30 | lapis block | 2 |
| spruce boat | 100 | iron pickaxe | 75 | diamond axe | 30 | slime | 2 |
| spruce door | 100 | iron boots | 75 | diamond pickaxe | 30 | dispenser | 1 |
| spruce fence | 100 | iron trapdoor | 75 | bookshelf | 25 | golden chestplate | 1 |
| spruce fence gate | 100 | carpet | 70 | banner | 25 | gold block | 1 |
| spruce stairs | 100 | bed | 70 | diamond boots | 25 | speckled melon | 1 |
| stick | 100 | mossy cobblestone | 70 | fishing rod | 25 | lead | 1 |
| stone | 100 | vine | 70 | piston | 25 | poisonous potato | 1 |
| stone axe | 100 | brick | 65 | compass | 20 | rabbit stew | 1 |
| stone brick stairs | 100 | clay ball | 65 | brick stairs | 20 | emerald block | 1 |
| stone button | 100 | leather chestplate | 65 | spider eye | 20 | enchanting table | 1 |
| stone hoe | 100 | bread | 65 | lapis lazuli | 20 | golden apple | 1 |
| stone pickaxe | 100 | chest minecart | 65 | glass | 20 | enchanted book | 0.5 |
| stone pressure plate | 100 | furnace minecart | 65 | sand | 20 | tnt | 0 |
| stone shovel | 100 | hopper | 65 | ink sac | 20 | tnt minecart | 0 |
| stone stairs | 100 | iron helmet | 65 | cooked rabbit | 20 | | |
| stone sword | 100 | minecart | 65 | rabbit | 20 | | |

