# OpenReview forum: "Ghost in the Minecraft: Hierarchical Agents for Minecraft via Large Language Models with Text-based Knowledge and Memory"
_ICLR.cc/2024/Conference — Submitted to ICLR 2024_

### Official Review · Reviewer_W7X3 · 2023-10-28

**Soundness:** 2 fair
**Presentation:** 3 good
**Contribution:** 2 fair
**Rating:** 3
**Confidence:** 5

**Summary:**

This paper proposes a LLM-based system to play video game Minecraft. It is hierarchical, relying on scripted behaviors for interaction with the game and using LLMs to accomplish high-level reasoning. The interfacing between LLMs and scripted actions is achieved through carefully designed action abstraction. Authors then demonstrate the performance of this system on collecting Minecraft items.

**Strengths:**

- The effort behind this submission to interface the Minecraft video game with text-based LLMs is impressive.
- The paper is well presented.

**Weaknesses:**

The biggest concern about this submission is limited technical novelty and scientific value.

First of all, using LLMs to assist high-level planning and reasoning in sequential decision-making tasks, such as those in Minecraft, is not novel (Huang et al., 2022ab, Wang et al., 2023ab, Yuan et al., 2023, Nottingham et al., 2023). While authors claim the proposed method is different because it is so-called *LLM-native*, I believe this should be more considered as a drawback, due to the fact that a huge amount of domain knowledge is required to carefully craft the action abstractions. Further, the proposed method emphasizes on using LLMs as planner and for reasoning, only few analysis, if any, is provided to explain why it is effective. This renders the proposed method more like an application of existing LLMs.

I do feel the scientific value is also limited due to inappropriate experiment settings. First of all, while authors claim the proposed system to address open-world tasks, only item collecting tasks are considered in the experiments. However, even in the Minecraft domain, possible task categories are not limited only to collecting items (Fan et al., 2022). To add on this point, the designed action abstraction overfits the fact that only this type of tasks is considered, further questioning the feasibility of this system to truly open-world tasks with diverse semantics. The main experimental comparison is also insufficient. Specifically, authors only formally compared against two methods, while one of them (AutoGPT) is even just an open-source software, not a proper academic paper. Although authors include results from RL-based methods such as Dreamer V3 and VPT, the distinct difference in settings makes them not directly comparable. While table 2 also shows learning steps for them, it would be more proper to add up the learning steps for the LLMs used in this system. Actually, it would be more appropriate to compare against Voyager (Wang et al., 2023b.)

Finally, the generality of the proposed system is also doubtful. Given that Minecraft is one of the most popular video games in the world, the LLM used in this system is very likely to be trained on related information. Therefore, it is unsure whether the performance gain is achieved by the proposed pipeline, or the fact that powerful LLMs know pretty well about the domain (Minecraft).

# References
- Huang et al., Language models as zero-shot planners: Extracting actionable knowledge for embodied agents, ICML 2022a.
- Huang et al., Inner monologue: Embodied reasoning through planning with language models, arXiv 2022b.
- Wang et al.,  Describe, explain, plan and select: Interactive planning with large language models enables open-world multi-task agents, NeurIPS 2023a.
- Wang et al., Voyager: An Open-Ended Embodied Agent with Large Language Models, arXiv 2023b.
- Yuan et al., Plan4mc: Skill reinforcement learning and planning for open-world Minecraft tasks, arXiv 2023.
- Nottingham et al., Do Embodied Agents Dream of Pixelated Sheep: Embodied Decision Making using Language Guided World Modelling, ICML 2023.
- Fan et al., MineDojo: Building Open-Ended Embodied Agents with Internet-Scale Knowledge, NeurIPS 2022.

**Questions:**

Inaccurate word usage. Need to clarify words including "physical interactions", "physical world", and "physical environment" that appear multiple times in the first section. Neither Minecraft itself is a physics simulation, nor agents play it through a physical interface (Baker et al., 2022).

---

### Official Review · Reviewer_iZAK · 2023-10-29

**Soundness:** 3 good
**Presentation:** 3 good
**Contribution:** 2 fair
**Rating:** 6
**Confidence:** 5

**Summary:**

GITM demonstrates the potential of LLMs at solving complicated open-world challenges in Minecraft.
It consists of a goal decomposer (LLM parser), LLM planner, feedback processing, and hand-coded LLM interface.

**Strengths:**

1. The concept and approach of utilizing Language Learning Models to tackle Minecraft, the best-selling game, is quite intriguing.
2. The author’s decision to manually program a low-level policy for MineDojo could have significant implications for future studies.
3. The authors do not seem to use any simplification to the environment.

**Weaknesses:**

1. The low-level policy requires hand-coding and does not generalize to any other games or more challenge combat tasks (ender dragon).
2. Requirement for interpreter/compiler feedback. This may not be possible for many legacy environments like Atari or real-world.
3. Marginal novelty. Lots of LLM agents, or even Minecraft agents of similar nature have been proposed.

**Questions:**

1. How does the LLM decomposer compare to rule-based search on the Minecraft Wiki page?

---

### Official Review · Reviewer_3ABz · 2023-10-31

**Soundness:** 3 good
**Presentation:** 3 good
**Contribution:** 2 fair
**Rating:** 3
**Confidence:** 5

**Summary:**

The paper addresses the need for adaptive agents in modern computer games, focusing on the expansive and dynamic open world of Minecraft. The authors introduce Ghost in the Minecraft (GITM), a novel hierarchical agent that integrates Large Language Models (LLMs) with text-based knowledge and memory. The agent is capable of constructing structured actions for interaction in Minecraft using textual descriptions. This hierarchical approach enables the decomposition of goals into sub-goals, actions, and operations, leveraging text knowledge and memory for improved performance and adaptability.

**Strengths:**

1. The paper is well-written and easy to understand.
2. The research conducted in this paper explores an intriguing field, and the application of LLM to an open world appears to be effective. However, the author's significant simplification of the environment raises concerns about the reliability of the results.

**Weaknesses:**

- Method Design:
    1. Memory: Will GITM be able to replicate its successful record in memory in the new environment? Since GITM uses a rule-based controller, it can execute any theoretically feasible plan successfully. Therefore, conducting experiments under this setting would be meaningless.
    2. Know ledge Library：GITM relies on a manually designed knowledge library, which is costly due to the numerous items and corresponding rules in Minecraft. Additionally, not all environments have access to this knowledge beforehand. As a result, GITM's ability to transition to other environments is significantly limited.
    3. Action API: GITM did not utilize an action space similar to that of minedojo or humans. Instead, it relied on manually designing numerous functions, which is both expensive and difficult to apply in different environments.
- Experimental Details
    1. Ablation on Language Models. The main component of GITM is a pretrained language model (LM). Therefore, the author should perform ablation experiments on the LM, using open-source models like LLaMA2.
    2. The feedback provided by human-writting during execution is much more detailed than what is available in the original Minecraft and other environments. Can GITM works when no human-written feedbacks are supported.
    3. To showcase the effectiveness and robustness of GITM, it is recommended to use more open-ended environments.
    4. What is the maximum time allotted for each task? Have there been any changes in the environment rules? What is the initial state of the agent upon birth? How many seeds does the environment generate? The author has overlooked several important details, which greatly impacts the reliability of experimental results.
- Environmental Settings
    1. Comparing the success rate of GITM in completing tasks like Minecraft with models such as VPT and Dreamer is unfair because these models are not intended for multi-tasking. Moreover, DEPS is an unsuitable comparison object due to its use of a learning-based control policy instead of the manually designed rule-based control policy used in GITM. Additionally, DEPS does not utilize privileged information like lidar, which is considered important in minedojo. The author should consider comparing with these methods under same setting that does not employ this information at least.
    2. The recent work Voyager utilized settings similar to GITM. For instance, it employed mineflayer as a control mechanism and translated the surrounding environment into a text-world format.I observed that the author utilized information from a 10*10 block area surrounding the agent. This undermines Minecraft's partial observation feature, which allows the agent to see information about items obstructed by blocks. Consequently, GITM achieved an abnormally high diamond acquisition rate. However, this practice is not permitted in Minecraft, even for human players.

    In summary, if GITM can only solve text-world and attempts to ignore the partial visual observation and control difficulty in Minecraft, I recommend the authors to use `crafter` or other similar textworlds instead of conducting experiments in Minecraft, as this is unacceptable in the community's perspective.

- Others
    1. Are GITM executation videos are available?
    2. It appears that GITM contains numerous details and has implemented significant changes to the environment, including modifications to the action space. The author can enhance this process by submitting code to improve transparency and reliability.

**Questions:**

See in weakness.

---

### Meta-Review · Area_Chair_T1TW · 2023-12-06

**Metareview:**

This paper proposes an LLM-based agent with memory for hierarchical control on a limited set of Minecraft tasks. While the work focuses on an exciting application of LLMs, the reviewers generally agreed that this largely empirical work lacks a rigorous experimental study. The experiments fail to compare to relevant baselines like Voyager, while comparing to previously published results for methods operating in drastically different problem settings. Moreover, as some reviewers pointed out, the approach taken in this work relies on access to highly-privileged information that other methods typically do not rely on, making the evaluation performance here hard to contextualize to other state-of-the-art LLM-based methods for Minecraft.

**Justification For Why Not Higher Score:**

Given many recent works have achieved state-of-the-art performance on Minecraft by leveraging LLMs, it is important for this work to compare to these prior works to highlight the novel scientific value of this study. However, such comparisons and detailed discussions are notably missing from this work, making it impossible to contextualize the additional scientific value of this work.

**Justification For Why Not Lower Score:**

N/A

---

### Decision · Program_Chairs · 2024-01-16

Reject